# Lightweight Tunnel Obstacle Detection Based on Improved YOLOv5

**DOI:** 10.3390/s24020395

**Published:** 2024-01-09

**Authors:** Yingjie Li, Chuanyi Ma, Liping Li, Rui Wang, Zhihui Liu, Zizheng Sun

**Affiliations:** 1Key Laboratory of Computing Power Network and Information Security, Ministry of Education, Shandong Computer Science Center (National Supercomputer Center in Jinan), Qilu University of Technology (Shandong Academy of Sciences), Jinan 250353, China; 10431210710@stu.qlu.edu.cn (Y.L.); 10431210576@stu.qlu.edu.cn (R.W.); 2Shandong High-Speed Group Co., Ltd., Jinan 250014, China; machuanyi2006@163.com; 3School of Qilu Transportation, Shandong University, Jinan 250100, China; yuliyangfan@163.com (L.L.); zhihui.liu@126.com (Z.L.); 4Jiangsu XCMG State Key Laboratory Technology Co., Ltd., Xuzhou 221004, China

**Keywords:** object detection, lightweight model, improved YOLOv5

## Abstract

Considering the high incidence of accidents at tunnel construction sites, using robots to replace humans in hazardous tasks can effectively safeguard their lives. However, most robots currently used in this field require manual control and lack autonomous obstacle avoidance capability. To address these issues, we propose a lightweight model based on an improved version of YOLOv5 for obstacle detection. Firstly, to enhance detection speed and reduce computational load, we modify the backbone network to the lightweight Shufflenet v2. Secondly, we introduce a coordinate attention mechanism to enhance the network’s ability to learn feature representations. Subsequently, we replace the neck convolution block with GSConv to improve the model’s efficiency. Finally, we modify the model’s upsampling method to further enhance detection accuracy. Through comparative experiments on the model, the results demonstrate that our approach achieves an approximately 37% increase in detection speed with a minimal accuracy reduction of 1.5%. The frame rate has improved by about 54%, the parameter count has decreased by approximately 74%, and the model size has decreased by 2.5 MB. The experimental results indicate that our method can reduce hardware requirements for the model, striking a balance between detection speed and accuracy.

## 1. Introduction

With the rapid development of traffic engineering in China, an increasing number of tunnels have been constructed in complex geological settings [1]. However, the limitations of manual tunnel construction and inspection work have gradually become apparent. The harsh environment, heavy workload, and safety hazards at tunnel construction sites make it unsuitable for workers to engage in prolonged activities. Using robots in tunnel environments to replace workers in performing hazardous and challenging tasks can effectively enhance work efficiency and ensure personnel safety.

Due to the complex environment of tunnel construction sites, various obstacles such as workers, construction vehicles, and traffic cones are present. When robots navigate in such environments, collisions with these obstacles can occur, leading to accidents. Therefore, it is necessary for robots to have autonomous obstacle avoidance capabilities. To achieve autonomous obstacle avoidance, real-time and high-precision detection and recognition of obstacles are required.

Traditional obstacle detection generally relies on sensors. Currently, commonly used sensors include ultrasonic sensors, lidar sensors, infrared sensors, etc. [2]. Ultrasonic ranging sensors [3] have a limited detection range and can only serve as auxiliary obstacle detection sensors. Although lidar sensors [4] have high detection accuracy, their detection range is limited. To cover the entire body of the device, multiple lidar sensors must be installed on the device, significantly increasing the cost of the robot. The working principle of an infrared sensor is similar to that of ultrasonic sensors and laser radar. Its advantages include low power consumption, compact size, and versatile applications. However, it also has drawbacks such as low resolution and a relatively short operating distance.

Given the limitations of traditional approaches, utilizing object detection algorithms for obstacle detection in tunnel scenarios not only reduces detection costs but also ensures accuracy while swiftly identifying obstacles at a distance. In this paper, we have proposed an improved YOLOv5n model that can run smoothly on low-power embedded devices and detect obstacles in tunnel scenarios with high accuracy. Compared to two-stage object detection algorithms, the YOLO series exhibits more efficient detection performance. Furthermore, YOLOv5 has seen widespread engineering applications, and the YOLOv5n variant adopts a lighter design, making it better suited for the detection requirements in our study’s scenario.

However, obstacle detection in tunnel construction scenarios currently still faces several key challenges, including: There is a lack of relevant research and a lack of publicly available datasets;The environment of a tunnel construction site is dim, making it difficult to identify detection targets;Robots have limited computing power and storage capacity, making it difficult to run larger models.

To address these challenges, this paper focuses on the following tasks: a dataset of obstacles in tunnel construction environments is constructed; various lightweight backbone networks are compared horizontally, and the optimal backbone network is selected to improve the frame rate; and adjustments are made to activation functions, attention mechanisms, neck convolution blocks, and upsampling method to restore detection accuracy. Compared to the original model, the proposed model significantly improves detection speed while minimizing the impact on accuracy. It can meet the real-time detection requirements of obstacles in tunnel scenarios.

## 2. Related Work

The environmental perception system is a critical component for enabling autonomy in robots. It utilizes various sensors to gather environmental information and transmits these data to the robot. Simultaneously, the environmental perception system places a high demand on the accuracy and comprehensiveness of environmental information, as this directly influences the robot’s ability to effectively complete tasks [5]. Within this system, the efficient implementation of obstacle detection is paramount.

Currently, obstacle detection for mobile robots relies primarily on the integration of various sensors and associated algorithms onboard the robot. Commonly used sensors include ultrasonic sensors, color cameras, and laser rangefinders [6,7]. Numerous studies have also demonstrated the effectiveness of traditional methods. Researchers like Cai Zixing [8] and others have worked on converting laser rangefinder coordinates into world coordinates, filtering the data, and creating grid maps to identify dynamic obstacles, effectively detecting road obstacles. Cheng Jian and colleagues [9] introduced a local optimal segmentation method based on point cloud gradients to address the issue of under-segmentation during the segmentation process. However, this method has complex calculations and relatively poor real-time performance. By using clustering algorithms on obstacle information, they can efficiently extract obstacle contours. The drawback is that when obstacles are far away, the algorithm may experience missed detections and over-segmentation. Wang Zhu [10] proposed an obstacle detection algorithm that uses laser rangefinders to segment clusters. This algorithm analyzes the data collected by laser rangefinders from different obstacles in different environments, taking into account the differences in data sparsity for various obstacles at varying distances. By combining the advantages of clustering algorithms, it improves obstacle segmentation and achieves high accuracy.

While relying on sensors and associated algorithms enables obstacle detection, there are still several drawbacks. These include increased additional costs, computational complexity, poor real-time performance, and the potential for missed detections, particularly when obstacles are at a greater distance.

With the continuous improvement of computing power and the rapid development of deep learning techniques, more and more researchers have started to utilize deep learning for obstacle detection. Deep learning-based object detection algorithms obtain training weights by inputting a large amount of data and undergoing repeated training. This process results in more accurate detection results as well as strong adaptability and robustness. Hao [11] proposed a deep learning-based approach for highway obstacle detection, using Mask R-CNN for both detection and classification of obstacles on the road. This method effectively identifies and marks the position of obstacles on the road. Zheng [12] introduced an improved YOLOv4 algorithm for detecting target obstacles. By modifying the network architecture of YOLOv4 and incorporating the Mobilenet network, this method achieves a balance between high detection accuracy and reduced computational complexity. Guan [13] presented an approach that combines deep learning with sensors for obstacle detection. By using an instance segmentation algorithm to extract feature regions of railway tracks, this method combines visual sensors and lidar to acquire data. Da [14] proposed a new network architecture based on deep learning algorithms which can use deep features to identify different obstacles, and its detection accuracy is very good. Liu et al. [15] improved the SSD deep learning object detection method, which improved the recognition effect of pedestrian obstacles in the complex environment of orchards. Li et al. [16] proposed an improved lightweight object detection method based on YOLOv3 for the identification of typical obstacles in orchards, including people, cement pillars, utility poles, etc.

Leveraging deep learning for obstacle detection brings numerous advantages, yet challenges persist in its application for obstacle detection in mobile robots, including high model complexity, large parameter counts, deployment difficulties, and slower detection speeds. To address these issues, when performing detection tasks on embedded devices such as robots, it is necessary to apply lightweight processing to the models. As the demand for lightweight models continues to increase, there has been a growing body of research on model lightweighting techniques. Xia et al. [17] proposed a lightweight object detection method that integrates Shufflenet v2 [18] with YOLOv5, achieving both lightweight and accurate object detection and recognition tasks. Luo et al. [19] employed Mobilenet v1 [20] to lighten the backbone network, improved the feature pyramid with adaptive spatial feature fusion [21] to enhance accuracy, and modified the loss function for improved reliability, achieving real-time detection on embedded devices. Yang et al. [22] introduced an improved lightweight safety helmet detection algorithm, YOLO-M3, by replacing the backbone network of YOLOv5s with Mobilenet v3 [23] for feature extraction, reducing the model’s parameters and computational complexity to meet the deployment requirements on embedded systems.

It can be concluded from the above that using deep learning technology for detection and recognition in various scenarios has become a mainstream trend. Due to the complex environment of tunnel scenes, obstacle target detectors used for training in natural scenes cannot be directly applied to tunnel scenes. Therefore, constructing obstacle detection methods with high detection efficiency for complex tunnel environments has in-depth research value.

## 3. The EnlightenGAN Network for Low-Light Enhancement

In tunnel construction scenarios, insufficient lighting is a common issue, and even with additional lighting measures, captured images remain dim and blurry, adversely affecting model training and detection performance. Therefore, it is necessary to enhance the acquired original images.

In this study, we employ a low-light enhancement image preprocessing method based on the EnlightenGAN network [24] to address this issue. Specifically, we apply low-light enhancement to the images, improving their exposure, with the goal of enhancing the accuracy of tunnel obstacle detection under low-light conditions.

Traditional image enhancement methods often require paired low-light and normal-light images for training. In contrast, EnlightenGAN can be trained in an efficient unsupervised manner, even in the absence of paired low-light and normal-light image sets. EnlightenGAN consists of two components: a generator network and a discriminator network.

The generator of EnlightenGAN performs image generation and enhancement tasks using the U-Net architecture. The generator takes dimly lit images and their corresponding attention maps as input, where these attention maps are generated based on the illumination intensity of the input images. Following the U-Net architecture, the EnlightenGAN network initially generates a set of feature maps with different scales. It then adjusts the size of the attention maps to match each feature map. Subsequently, the adjusted attention maps are element-wise multiplied with all intermediate feature maps and subjected to addition operations, resulting in the final enhanced image.

This design effectively leverages the autoencoder structure of U-Net, allowing the transmission of information from the input image to various feature layers. Combined with the attention mechanism, the generator can selectively enhance image characteristics such as brightness, contrast, and color saturation. Therefore, EnlightenGAN is capable of significantly improving image quality without the need for paired datasets. Figure 1 illustrates the generator network structure of EnlightenGAN.

The discriminator of EnlightenGAN comprises a global discriminator and a local discriminator, both working in tandem to handle spatial variations in lighting conditions within images. When an image is input into the discriminator network, the global discriminator evaluates the entire image to determine its overall quality and authenticity. In contrast, the local discriminator uses randomly cropped small patches from the image to assess various aspects of the image, ensuring that the generated image maintains high quality in terms of local details.

This dual discriminator design allows for a more comprehensive evaluation of the generated images by considering both global and local factors. This design contributes to the improvement of image quality, ensuring that the generated images can compete with real images in various aspects. Figure 2 illustrates the discriminator network structure of the model.

## 4. Model

For detection tasks in tunnel construction scenarios, it is essential to achieve real-time detection while maintaining accuracy. This necessitates lightweight optimization of the model, striking a balance between ensuring accuracy and avoiding significant accuracy reduction. Therefore, this paper proposes a series of model improvements to make it more suitable for deployment on embedded devices.

### 4.1. Model Selection

Deep learning-based object detection algorithms can be divided into two categories: two-stage object detection and one-stage object detection. Two-stage object detection algorithms include R-CNN [25], Fast R-CNN [26], and Faster R-CNN [27]. They have achieved remarkable detection accuracy. However, due to their drawbacks, such as high computational cost for region proposal and slow detection speed, they cannot meet the requirements of real-time detection.

In contrast, the YOLO series [28,29] algorithms offer the advantages of high detection accuracy and fast detection speed. YOLO adopts the concept of one-stage object detection, which predicts the class and location information of objects end-to-end using a convolutional neural network. With the advancement of deep learning and algorithm improvements, YOLO has evolved from its initial version (v1) to the latest version (v8).

Considering practical engineering applications and algorithm lightweight requirements, YOLOv5 is better suited to meet the real-time detection requirements of robots in complex tunnel environments. In this paper, YOLOv5-6.0 is used as the basic unit of our model.

The network architecture of YOLOv5-6.0 is illustrated in Figure 3. The backbone network, known as CSPDarknet53, is responsible for extracting features from input images, capturing essential semantic information. To further enhance feature extraction and detection accuracy, it utilizes a neck network that integrates multi-scale features from different layers of the backbone network. This allows the model to capture both fine-grained details and high-level contextual information. It also employs multiple detection heads, each responsible for predicting object bounding boxes and class labels, ensuring accurate and efficient object detection. By combining these components, YOLOv5-6.0 achieves outstanding performance in object detection tasks, striking a balance between accuracy and computational efficiency.

YOLOv5-6.0 includes five different models, with varying sizes from smallest to largest: YOLOv5n, YOLOv5s, YOLOv5m, YOLOv5l, and YOLOv5x. The official performance comparison of various versions of YOLOv5 on the COCO public dataset is shown in Table 1. Although YOLOv5x achieves the best detection performance, its computational complexity is high due to its deep network architecture, resulting in slow inference speed. In contrast, lightweight models are more suitable for porting to embedded devices for industrial applications. Therefore, in this paper, we selected YOLOv5n as the base model.

### 4.2. Modifying the Backbone Network

The detection scenario in this paper is the tunnel construction environment, and the model needs to be deployed on embedded devices for detection. Due to the limited computational resources and storage capacity of embedded devices, it is necessary to lightweight the model to ensure efficient operation.

We replaced the backbone network part of the original model with various lightweight backbone networks, including MobileNet v3, Ghostnet [30], and Shufflenet v2. These networks were individually trained on the dataset constructed in this paper, and the lightweighting effects of different networks on the model were compared. The performance comparison results between different backbone networks are shown in Table 2.

By considering both detection accuracy and speed, the YOLOv5n backbone network was replaced with Shufflenet v2, which was designed and proposed by Ma et al. In the original Shufflenet v1 structure, a large number of group convolution operations are used, which reduce computational costs but increase memory access costs, resulting in slower model execution speed. In Shufflenet v2, the authors introduced a Channel Split operation to divide the input channels into two branches, avoiding the use of group convolution operations. The two branches are then fused together using Concat and Channel Shuffle to ensure information exchange. The network structure of Shufflenet v2 is shown in Figure 4, where Figure 4a,b represent the basic module and downsampling module of Shufflenet v2, respectively.

In the basic module shown in Figure 4a, the input feature channels are split into two branches using the Channel Split operation. One branch remains unchanged, while the other branch sequentially performs three operations with a stride of 1 each, including a pointwise convolution, a depth-wise separable convolution, and another pointwise convolution. The number of input and output channels remains the same for each branch. The outputs of the two branches are then concatenated using the Concat operation, and finally, the Channel Shuffle operation is applied to fuse the features between the two branches.

As shown in Figure 4b, the input is split into two branches. The left branch undergoes a depth-wise separable convolution and a pointwise convolution, both with a stride of 2. The right branch follows the same operations as the basic module but with a stride of 2. The outputs of the two branches are then concatenated, resulting in a feature map with doubled channels and spatial dimensions reduced by half. Finally, the Channel Shuffle operation is applied to fuse the features between different groups.

To enhance detection accuracy and achieve model lightweighting, we used an alternating stacking arrangement of the basic module and downsampling module of Shufflenet v2 to replace the C3 and Conv modules, respectively. Furthermore, the *ReLU* activation function in the Shufflenet v2 module was replaced with the *SiLU* [31] activation function. The formulas for *ReLU* and *SiLU* are shown in Equations (1) and (2).
(1)$$ReLU=\left\{\begin{array}{c} x,x>0\\ 0,x\leq 0 \end{array}\right.$$
(2)$$SiLU=\frac{x}{1+e^{-x}}$$

Although the *SiLU* function introduces slightly more computation compared to the *ReLU* function, it addresses the issue of neuron death caused by the *ReLU* function’s gradient being zero for *x* < 0. By replacing the *ReLU* function with the *SiLU* function, the network can retain the advantages of *ReLU* while better handling the neuron death problem. This allows for adjustments to the model architecture, reducing the number of parameters and computational load without sacrificing network performance.

Furthermore, a designed module, i.e., CBSM, is introduced in the first layer of the backbone network, which performs a series of operations, including convolution, batch normalization, max pooling, and activation, on the input image. The specific structure of CBSM is shown in Figure 5.

The functionality of CBSM is consistent with the first layer of the original YOLOv5n model. However, CBSM has significantly fewer parameters, with only 232 parameters compared to the original model’s first layer with 1760 parameters. This reduction in computational complexity while maintaining the same functionality makes it more suitable for embedded devices.

The modified backbone network structure is illustrated in Figure 6.

### 4.3. Attention Module

The detection task in this paper involves multiple classes of objects, and under low-light or complex lighting conditions, some targets may be more challenging to identify. Therefore, we introduced an attention mechanism to allow the model to focus more on essential target features, thereby improving the detection accuracy for different categories of targets.

We incorporated multiple attention mechanisms into the last layer of the backbone network and conducted training on the dataset established in this study. The aim was to assess and compare the comprehensive performance of various attention mechanisms. These mechanisms include channel attention mechanisms such as squeeze-and-excitation networks (SE) [32] and efficient channel attention (ECA) [33], and hybrid attention mechanisms that combine channel attention and spatial attention, such as the convolutional block attention module (CBAM) [34] and coordinate attention (CA) [35], as well as the C3_CA module obtained by combining the C3 module with the CA attention mechanism. Table 3 shows the comparison of adding various attention mechanisms at the same location, specifically the final layer of the model’s backbone network. After comparison, we chose to incorporate the C3_CA module with the aim of maximizing the recovery of accuracy loss resulting from the change in the backbone network. This module combines the feature enhancement capability of C3 with the positional awareness capability of CA, enabling the model to focus more on relevant key information and reduce interference from background information, thus enhancing overall performance.

The coordinate attention mechanism aims to capture attention along the horizontal and vertical directions of the feature map and encode precise positional information. It consists of two steps: coordinate information embedding and coordinate attention generation.

The coordinate information embedding step involves dividing the input feature map into horizontal and vertical directions and performing average pooling to obtain two one-dimensional directional awareness feature maps. The process can be represented by the following equations:(3)$$z_{c}^{h}\left(h\right)=\frac{1}{W}\sum_{i=0}^{W}x_{c}\left(h,i\right)$$
(4)$$z_{c}^{w}\left(w\right)=\frac{1}{H}\sum_{j=0}^{H}x_{c}\left(j,w\right)$$

The coordinate attention generation step involves concatenating the two embedded feature maps, which contain directional information, and passing them through convolutional, batch normalization, and *SiLU* activation modules to encode spatial information and obtain a dimensionally reduced feature map. This process can be shown in Equation (5), where $\left[z^{h},z^{w}\right]$ represents the concatenation of the two embedded feature maps, *F*_1_ ($\left[z^{h},z^{w}\right]$) denotes the batch normalization operation applied to the concatenated feature maps, δ represents the *SiLU* activation function, and *f* represents the resulting feature map after the coordinate attention generation.
(5)$$f=\delta \left(F_{1}\left(\left[z^{h},z^{w}\right]\right)\right)$$

The next step is to decompose the obtained feature map ‘f’ into horizontal and vertical attention tensors by splitting it along the spatial dimension. Each tensor is then passed through a convolutional layer to increase its dimension, resulting in the feature maps ${‘F}_{h}’$ and $‘F_{w}’$, which have the same size as the input feature map. Sigmoid is utilized to obtain the attention weights ${‘g}^{h}’$ and $‘g^{w}’$. The process is shown as follows:(6)$$g^{h}=\sigma \left(F_{h}\left(f^{h}\right)\right)$$
(7)$$g^{w}=\sigma \left(F_{w}\left(f^{w}\right)\right)$$

The attention mechanism focuses on key information in the feature map by performing element-wise multiplication with the input feature map. The final output is shown in Equation (8). The structure of the coordinate attention mechanism is shown in Figure 7.
(8)$$y_{c}\left(i,j\right)=x_{c}\left(i,j\right)\times g_{c}^{h}\left(i\right)\times g_{c}^{w}\left(j\right)$$

The C3 module is composed of multiple convolutional modules and a BottleNeck module, divided into two branches for processing. It enhances the expressive power of features by adjusting the channel number and depth of the feature maps. In one branch, the feature map undergoes a 1 × 1 convolution operation to reduce the channel number before entering the BottleNeck module. This design aims to facilitate a deeper understanding and representation of feature information by adjusting the channel number. The feature map in the other branch undergoes only a 1 × 1 convolution operation. Finally, the feature maps from both branches are fused through the Concat operation and integrated overall through an additional 1 × 1 convolutional layer.

We introduced the coordinate attention (CA) mechanism into the BottleNeck module of the C3 module, combining CA with C3 to form the C3_CA module. The network structure of C3_CA is illustrated in Figure 8.

By placing C3_CA at the last layer of the improved model’s backbone network, we enhanced the network’s ability to extract crucial information. Inserting the attention module at this position effectively controls computational complexity, ensuring the timeliness of detection.

The backbone network structure diagram with the added attention mechanism is shown in Figure 9.

### 4.4. Improved Neck Convolution Block

Although the backbone network has been replaced with Shufflenet v2, there are still a large number of convolutional layer convolution operations which consume more resources. To better capture both local and global information in the tunnel construction environment and strike a balance between accuracy and inference time, we made modifications to the model’s neck network to further enhance detection performance. Li et al. [36] proposed a new convolution method called GSConv, as they found that networks constructed with depth-wise separable convolutions have poor accuracy.

In this paper, all standard convolutional modules in the neck region are replaced with GSConv convolution to improve the model’s accuracy. Although Shufflenet v2 significantly improves detection speed through depth-wise separable convolution, the channel information in depth-wise separable convolution is separated during the computation process, resulting in lower feature extraction and fusion capabilities and a significant decrease in accuracy.

GSConv combines depth-wise separable convolution with standard convolution. It concatenates the results of depth-wise separable convolution and standard convolution, combining their feature maps. Then, the Shuffle operation is applied to achieve mutual permutation between the feature maps from depth-wise separable convolution and standard convolution. While depth-wise separable convolution almost completely severs the connections between each channel, standard convolution can effectively preserve these connections. Therefore, GSConv can alleviate model complexity while maintaining accuracy, striking a balance between accuracy and inference time. The network structure of GSConv is illustrated in Figure 10. The modified neck network structure is illustrated in Figure 11.

### 4.5. CARAFE Upsampling Operator

In the context of object detection tasks, it is crucial to ensure that the model can perform detection effectively across various scales and seamlessly integrate detection results. This necessitates the use of upsampling operations to adjust the size of input feature maps to match the dimensions of the original image, thus enabling the model to detect objects of various sizes and distances effectively. Traditional upsampling methods often rely on bilinear interpolation. However, these methods have inherent limitations that may result in the loss of crucial image details. Furthermore, the process of predicting upsampling kernels with traditional methods introduces a significant computational and parameter overhead, which is unfavorable for achieving lightweight network architectures.

To reduce the deviation between the model’s detection boxes and real targets without adding excessive computational complexity, we introduced the CARAFE [37] lightweight upsampling operator, replacing the previous upsampling module with CARAFE in the model. It can improve the performance of model upsampling, achieving more precise localization. CARAFE achieves an expanded receptive field and establishes correlations between input information and feature maps without introducing excessive computational complexity or additional parameters. This innovation leads to superior performance compared to traditional upsampling methods. CARAFE provides object detection tasks with enhanced feature representations and improved preservation of fine-grained details, ultimately boosting model efficiency and effectiveness.

The CARAFE operation divides the upsampling into two modules, namely the upsampling kernel prediction module and the content-aware reassembly module. In the upsampling kernel prediction module, for an input image of dimensions H × W × C, it begins by reducing the channel dimensions of the input feature map to H × W × C_m_ using a 1 × 1 convolution, thereby reducing computational complexity. Subsequently, for convolution encoding, channel redistribution is applied. This is followed by pixel rearrangement to enlarge the upsampling receptive field. Finally, a normalization step is performed to minimize parameter counts in the operation. In the Feature Recombination module, the feature maps obtained from the upsampling kernel prediction module are point-multiplied with those obtained from standard upsampling to produce the final output. The network structure of CARAFE is illustrated in Figure 12. The modified neck network structure is illustrated in Figure 13.

### 4.6. Framework of the Improved YOLOv5

YOLOv5 was improved by integrating the Shufflenet v2 network, attention mechanism, neck convolution block, and CARAFE upsampling operator. Replacing the backbone network with Shufflenet v2 enables fast feature extraction of input images at different scales. The introduction of the attention mechanism between the backbone network and the neck enhances the network’s ability to extract and locate crucial information. Replacing the convolution module in the neck network with the GSConv block and modifying the upsampling method to CARAFE further enhances the model’s accuracy. Additionally, due to the change in network depth, modifications were made to the corresponding parameters of concatenation operations and detection heads. The overall framework of the proposed model in this paper is illustrated in Figure 14.

## 5. Experiments

### 5.1. Dataset

Since there is no publicly available obstacle dataset specifically for tunnel construction scenarios, it was necessary to create a custom dataset. This involved capturing a large number of photos at tunnel construction sites and performing preprocessing and annotation on the images. The images in the dataset were captured using a Xiaomi 11 smartphone at an approximate height of 1.5 m. After preprocessing, the images had a resolution of 1504 × 1128.

Due to the complex environment at tunnel construction sites and the difficulty in data collection, it was necessary to augment the collected image data to increase the quantity and spatial diversity of the dataset. This includes applying techniques such as horizontal flipping and brightness enhancement to enhance the learning capability of the deep neural network. After data augmentation, a total of 8000 images were obtained. The dataset was categorized into six classes based on representative objects commonly found at the construction site: worker, construction vehicles, and traffic cones. The corresponding labels for these classes were assigned as “people”, “excavator”, “mixer truck”, “muck truck”, “shovel car”, and “traffic cone”, respectively. After annotation, the approximate number of labels for each category was as follows: 12,000 for “people”, 2100 for “excavator”, 1700 for “mixer truck”, 1750 for “muck truck”, 1600 for “shovel car”, and 4100 for “traffic cone”. Figure 15 shows the distribution of the number of labels in the dataset.

Due to the difficulty in collecting images in tunnel construction scenarios, this paper allocated as many images as possible to the training set to assist the model in better learning target features. The dataset was ultimately randomly divided into training, validation, and test sets in a ratio of 8:1:1. The training set was used to train the model parameters, the validation set was used to assess the model’s generalization ability, and the test set was used to evaluate the final model’s accuracy. This partitioning allowed for effective training, validation, and evaluation of the model, ensuring both its generalization ability and accuracy. Figure 16 shows representative images from different categories in the dataset.

### 5.2. Implementation Details

The training platform parameters of this model were as follows: the processor was an Intel (R) Xeon (R) Gold 6326 CPU; the graphics card was an NVIDIA GeForce RTX 3090, with 24 GB of graphics storage; the system environment was Ubuntu 20.04.5; the GPU acceleration environment was CUDA 11.3; the programming language version was Python 3.8; and the training framework was Python 1.10.0.

The training parameters of different models were consistent, and the training parameters were set as follows: based on the convergence observed during the training process on the dataset used in this paper, the maximum number of iteration epochs was set to 100; to balance memory efficiency and computational speed, the batch size was set to 32; pre-trained weights were not used during the training process to ensure that the model was specifically trained for our tunnel obstacle detection task; and the other super parameters remained on the default settings, that is, the initial learning rate was 0.01, the weight attenuation coefficient was 0.0005, the learning rate momentum was 0.937, and the random gradient descent was used as the optimizer.

### 5.3. Performance Metrics

In this paper, we employed *Recall* (R), *Precision* (P), and mean Average Precision (*mAP*) to evaluate the model performance. These metrics play a crucial role in assessing the model’s ability to correctly identify positive samples and avoid misclassifications, and overall detection accuracy. *Recall* measures the proportion of actual positive samples that are correctly predicted as positive, while *Precision* assesses the accuracy of positive predictions. *mAP*, as the mean of Average Precision values across all classes, provides an aggregate measure of detection performance.

The calculation formulas for *Recall* and *Precision* are shown as Equations (9) and (10). In these equations, *TP* denotes true positives, which are positive samples correctly classified as positive by the model. *FN* represents false negatives, which are positive samples incorrectly classified as negative. *FP* stands for false positives, which are negative samples incorrectly classified as positive.
(9)$$Recall=\frac{TP}{TP+FN}$$
(10)$$Precision=\frac{TP}{TP+FP}$$

The curve plotted with *Precision* on the y-axis and *Recall* on the x-axis is commonly referred to as the P-R curve. The area under the P-R curve, enclosed by the curve and the coordinate axes, is known as Average Precision (*AP*). The mean of *AP* values across all classes is denoted as mean Average Precision (*mAP*). A higher value of *AP* and *mAP*, closer to 1, indicates more accurate detection performance by the model. The formulas for calculating *AP* and *mAP* are shown in Equations (11) and (12).
(11)$$AP=\int_{0}^{1}P\left(R\right)dR$$
(12)$$mAP=\frac{1}{n}\sum_{i=1}^{n}AP\left(i\right)$$

When evaluating object detection models, in addition to considering accuracy metrics, it is also necessary to take into account model complexity and detection speed. Model complexity is typically measured by metrics such as computational complexity and the number of parameters. Model detection speed is measured by *Frame rate*. The calculation of frame rate involves three components: preprocessing time (*T*_1_), inference time (*T*_2_), and non-maximum suppression time (*T*_3_). The total time taken for inference on a single image is the sum of these three components. The inverse of this total time gives the value of FPS, as shown in Equation (13).
(13)$$Frame\;rate=\frac{1}{T_{1}+T_{2}+T_{3}}$$

GFLOPs is a crucial metric for assessing model performance and computational overhead. It represents Giga Floating Point Operations Per Second, and a higher value generally indicates that the model performs more floating point operations during inference or training, signifying greater computational complexity.

### 5.4. Low-Light Image Enhancement

In tunnel environments, images are often compromised by low exposure, resulting in the loss of crucial feature information and thus increasing the complexity of model detection. Enhancing the original images through EnlightenGAN makes the images clearer and more suitable for model detection. Figure 17 illustrates a comparison between the enhanced and original images, as well as a comparison of their detection performance within the same model.

Through the enhancement by the EnlightenGAN network, there is a noticeable improvement in image quality. Comparative results before and after enhancement indicate that introducing EnlightenGAN enhances the network’s detection accuracy in low-light conditions, enabling the model to identify and detect objects that were previously challenging to recognize. Additionally, it helps reduce the model’s false positives and false negatives to some extent, thereby enhancing the model’s detection performance.

### 5.5. Obstacle Detection Experimental Results

Using the aforementioned improvement methods, this paper trained multiple models with the dataset constructed in this study and conducted experimental tests on the same device to evaluate the performance of different models. The improvements are categorized as follows: Point 1 involves replacing the original model’s backbone network with Shufflenet v2; Point 2, based on Point 1, introduces the C3_CA mechanism; Point 3, based on Point 2, replaces the neck convolution block with the GSConv module; and Point 4, based on Point 3, changes the upsampling method to CARAFE. The results of the obstacle detection experiments are presented in Table 4.

According to Table 4, while YOLOv7-tiny exhibits the highest detection accuracy, both its computational complexity and parameter count are significantly higher than YOLOv5n, making it unsuitable for deployment on embedded devices. On the other hand, although YOLOv5n is more suitable for mobile terminal usage compared to YOLOv7-tiny, there is still potential for further lightweight optimization. Therefore, we made additional modifications to its network. In Point 1, the lightweight Shufflenet v2 was employed to replace the backbone network, resulting in an approximately 51% increase in inference speed and nearly doubling the frame rate. However, this leads to a decrease in mAP of about 7.1%. As our goal is to achieve a balance between detection speed and accuracy, further improvements were made. In Point 2, the addition of attention mechanisms restored detection accuracy, increasing mAP by 5%, but with a more than 1.5-fold increase in the model’s parameter count. In Point 3, modifications to the model’s neck network led to a slight increase of 0.3% in accuracy, while reducing the parameter count by 0.03 M. After these adjustments, it was observed that the model exhibited some deviation between detection boxes and targets. Therefore, in Point 4, the CARAFE operator was introduced to enhance the performance of upsampling, improving precise target localization and further increasing the detection accuracy to 95.2%.

Ultimately, through this series of improvements, compared to the initial model, the proposed model, while experiencing a 1.5% decrease in accuracy, achieved a roughly 37% increase in inference speed, a 54% increase in frame rate, a reduction of 1.2 M in parameters, and a decrease of approximately 64% in model size. Considering the overall trade-off between detection accuracy and speed, the modified model is more suitable for deployment on embedded devices. The detection results are shown in Figure 18.

## 6. Conclusions

This paper proposes a lightweight obstacle detection model based on YOLOv5n specifically designed for tunnel construction scenarios. The primary objective of this model is to reduce the deployment cost of complex models while enhancing detection speed, achieving a balance between detection accuracy and speed.

Based on the above experimental data and analysis, our improved model demonstrates superior performance in terms of detection speed and model size, making it more suitable for deployment on embedded devices. Firstly, by introducing the Shufflenet v2 lightweight backbone network, we nearly doubled the model’s frame rate, significantly reducing computational complexity and the number of parameters, providing robust support for efficient model performance in practical applications. Secondly, the incorporation of the coordinate attention mechanism further enhanced object detection accuracy, addressing the challenges posed by complex tunnel environments and improving the model’s robustness in detection tasks. These improvements not only increased the frame rate while maintaining accuracy but also made the model better suited for the demands of real-world tunnel construction sites. Finally, adjustments to the model’s upsampling method and neck network further reduced computational costs while enhancing the network’s capabilities in feature extraction and fusion. Comparing our proposed improved model with the original model, our model achieved an approximately 37% increase in inference speed, a 54% increase in frame rate, a reduction of 1.2 million parameters, and a decrease in model size of approximately 64%.

In conclusion, our improved model achieves a balance between detection accuracy and speed, meeting the requirements of obstacle detection in tunnel construction scenarios. It provides a lightweight, efficient, and cost-effective solution for obstacle detection in tunnel construction environments.

## Figures and Tables

**Figure 1 sensors-24-00395-f001:**
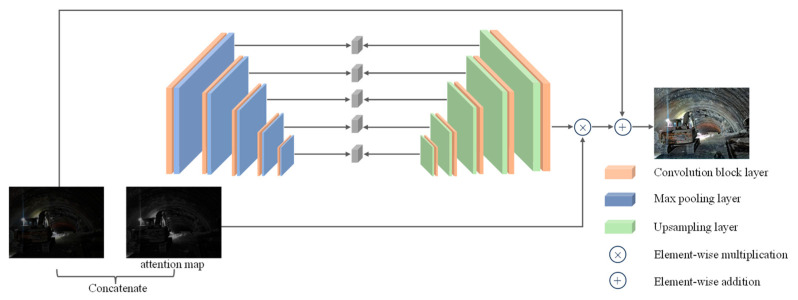
Generator network architecture of EnlightenGAN.

**Figure 2 sensors-24-00395-f002:**
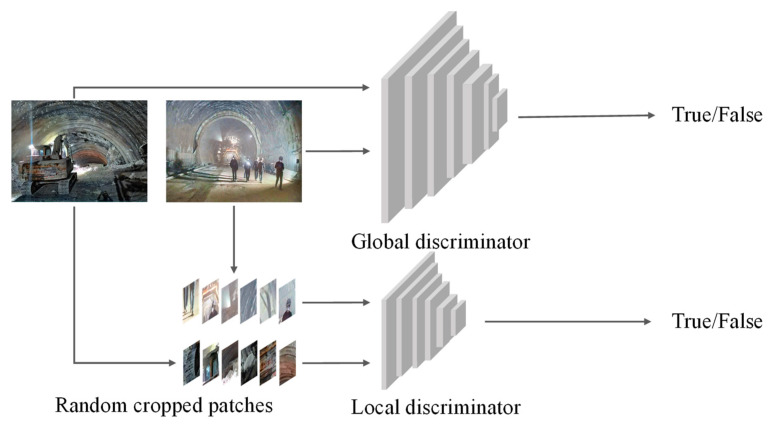
Discriminator network architecture of EnlightenGAN.

**Figure 3 sensors-24-00395-f003:**
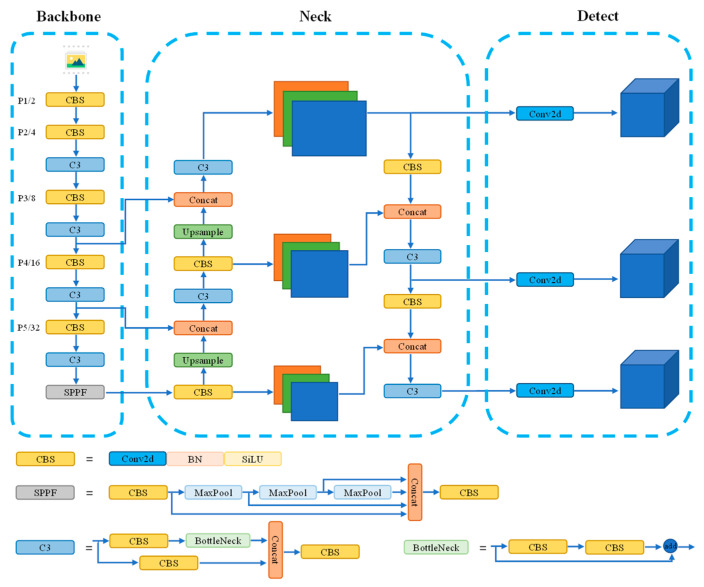
YOLOv5 network architecture. Different colored squares represent distinct modules, with each color indicating a specific type of operation and the three dashed boxes correspond to the three com-ponents of the YOLOv5 model. The straight arrows indicate the next layer’s output of the model.

**Figure 4 sensors-24-00395-f004:**
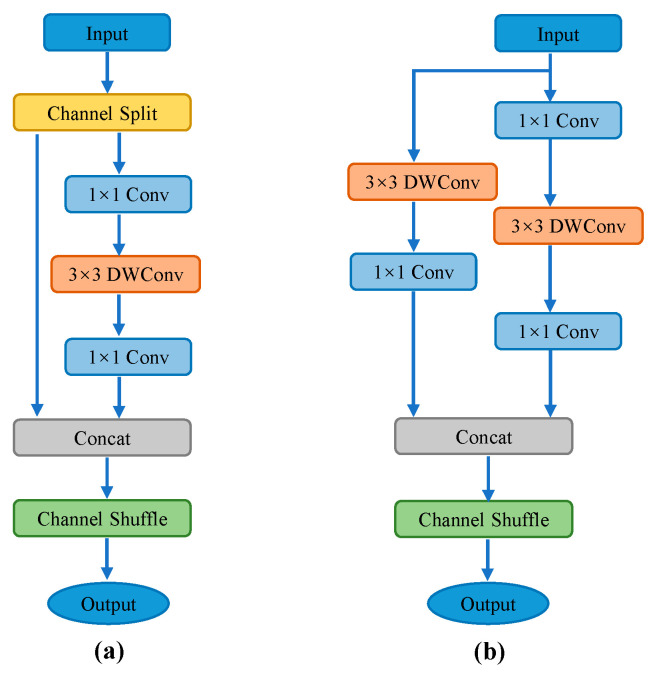
Shufflenet v2 network structure. (**a**,**b**) represent the basic module and downsampling module of Shufflenet v2, respectively.

**Figure 5 sensors-24-00395-f005:**
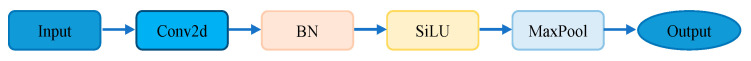
CBSM network structure.

**Figure 6 sensors-24-00395-f006:**
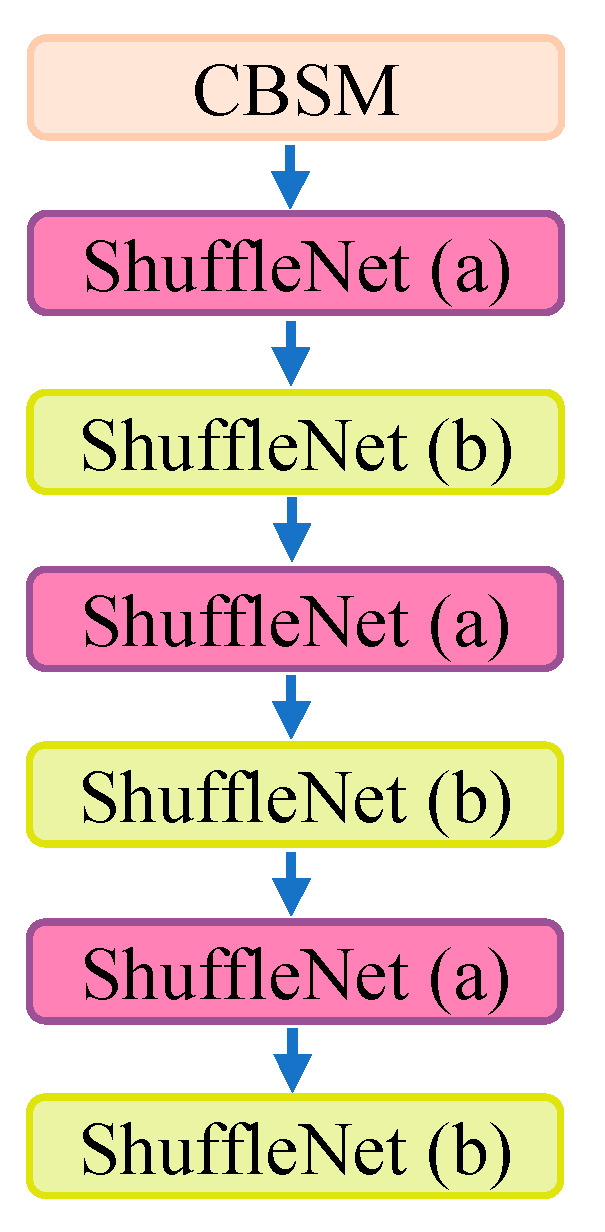
The modified backbone network structure. ShuffleNet (a) and ShuffleNet (b) represent the basic module and downsampling module of Shufflenet v2, respectively.

**Figure 7 sensors-24-00395-f007:**
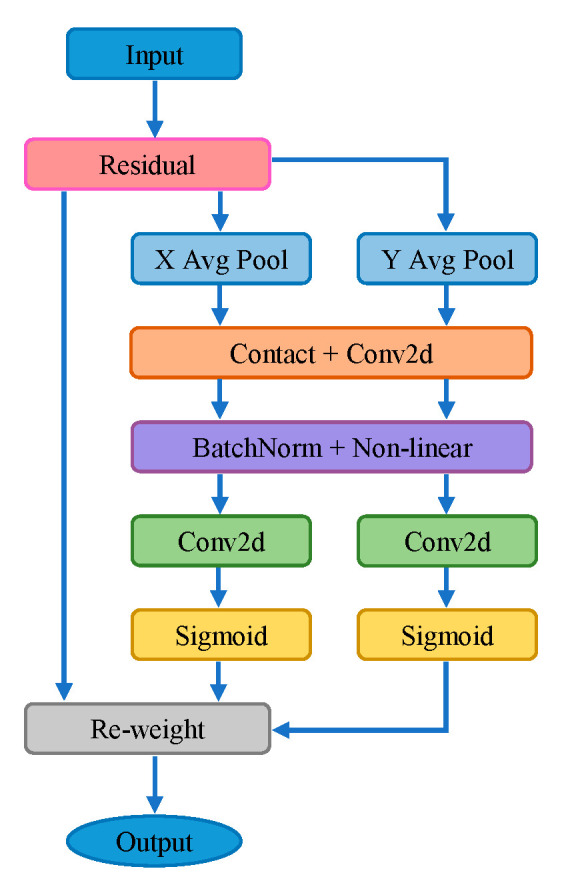
Coordinate attention module structure.

**Figure 8 sensors-24-00395-f008:**
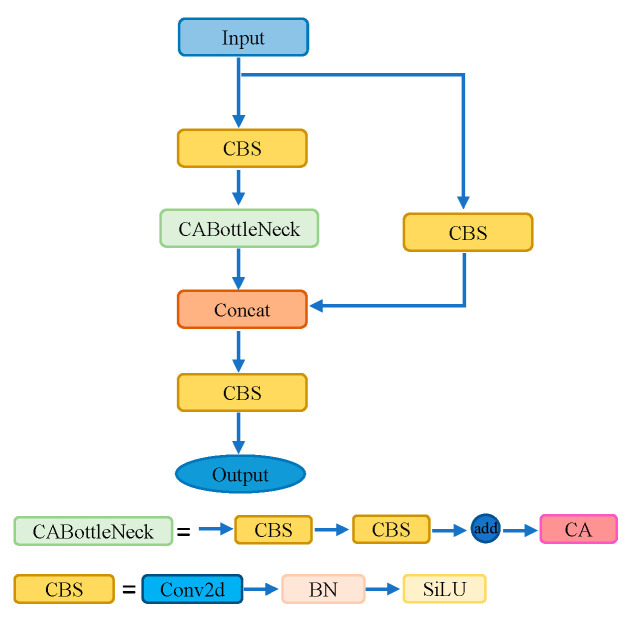
C3_CA module structure.

**Figure 9 sensors-24-00395-f009:**
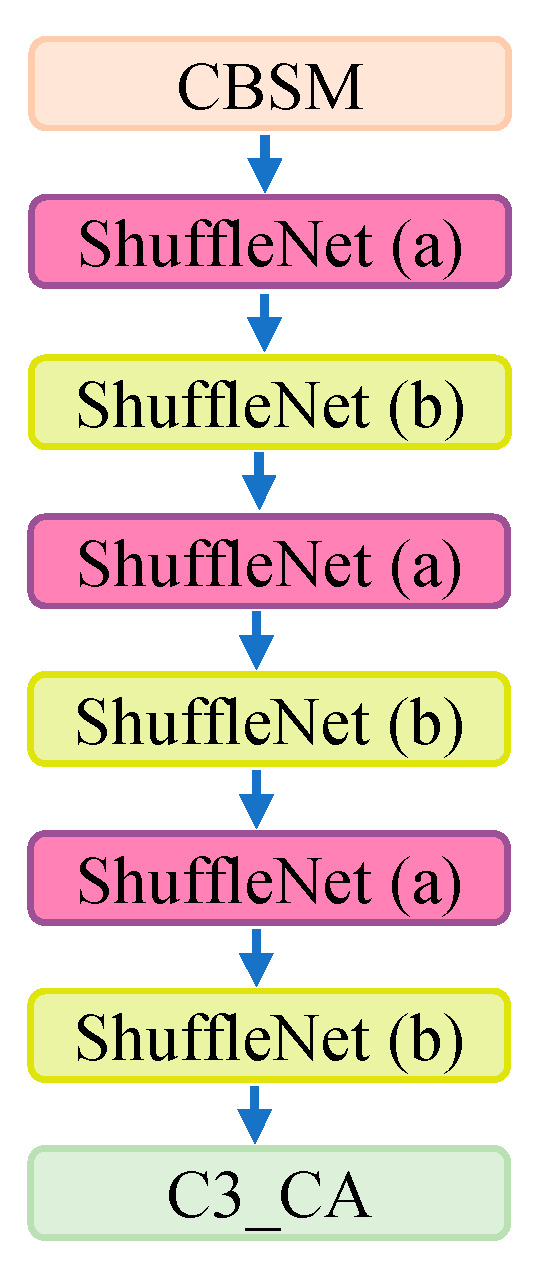
The modified backbone network structure with C3_CA. ShuffleNet (a) and ShuffleNet (b) represent the basic module and downsampling module of Shufflenet v2, respectively.

**Figure 10 sensors-24-00395-f010:**
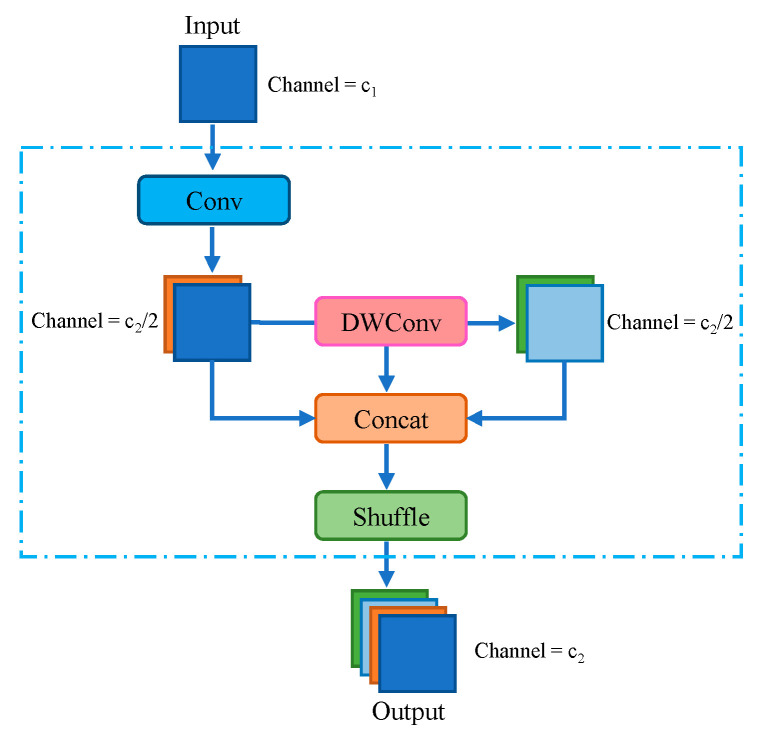
GSConv module structure. Different colored squares represent distinct modules, with each color indicating a specific type of operation.

**Figure 11 sensors-24-00395-f011:**
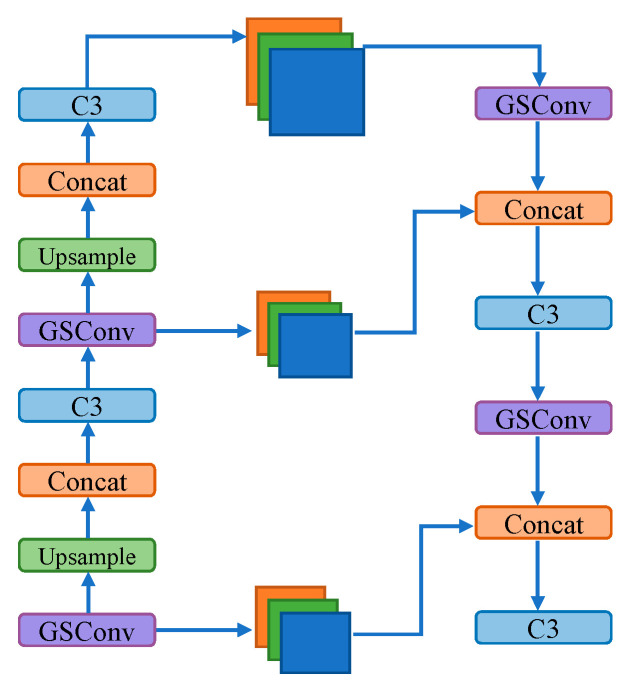
The modified neck network structure with GSConv.

**Figure 12 sensors-24-00395-f012:**
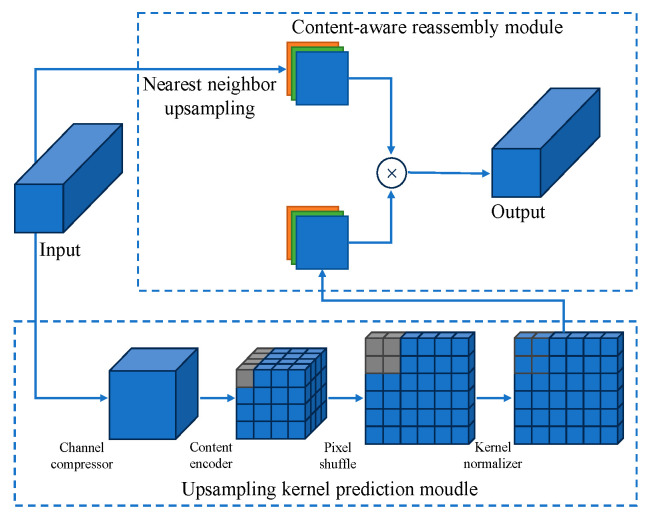
CARAFE module structure. The gray area represents the changes in the region during the operation.

**Figure 13 sensors-24-00395-f013:**
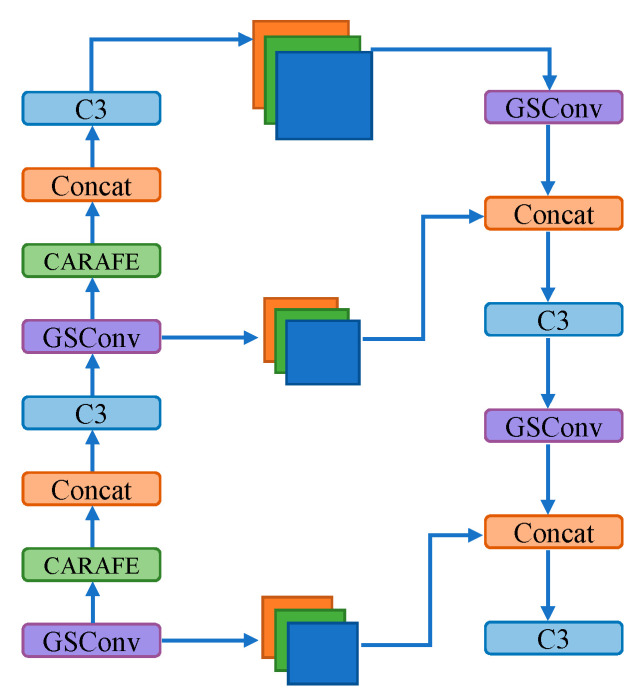
The modified neck network structure with CARAFE.

**Figure 14 sensors-24-00395-f014:**
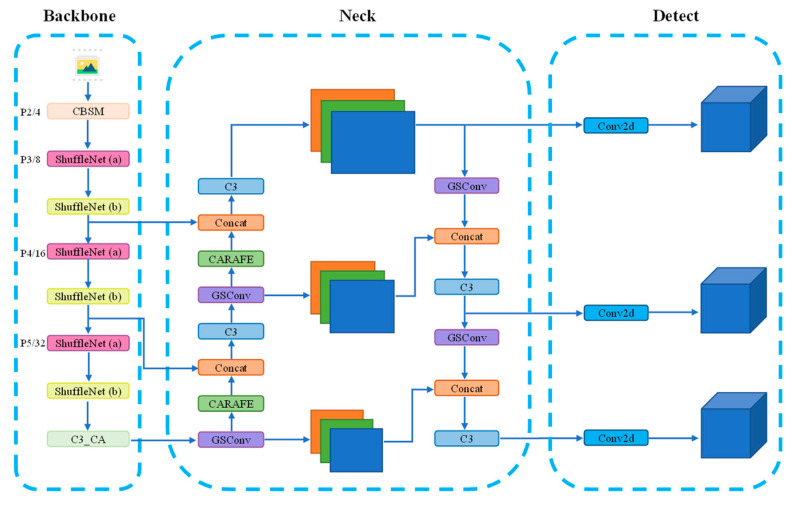
Improved YOLOv5 network structure. Different colored squares represent distinct modules, with each color indicating a specific type of operation. ShuffleNet (a) and ShuffleNet (b) represent the basic module and downsampling module of Shufflenet v2, respectively.

**Figure 15 sensors-24-00395-f015:**
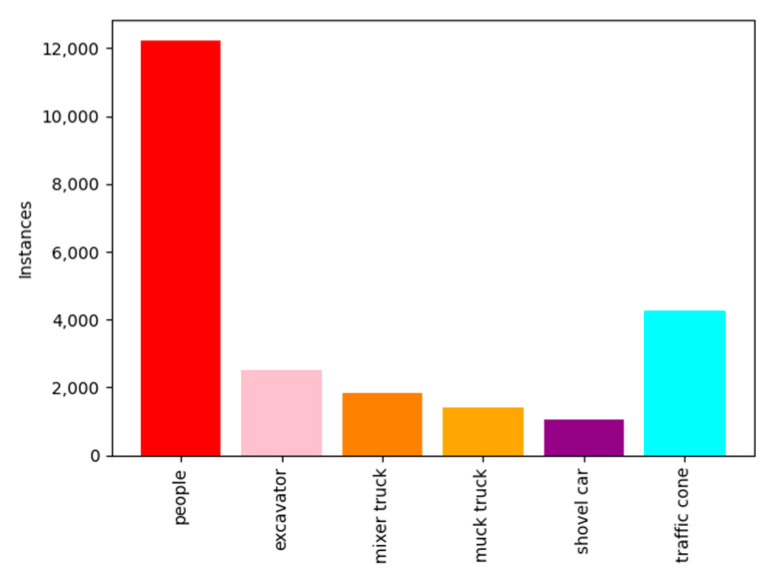
The distribution of the number of labels in dataset.

**Figure 16 sensors-24-00395-f016:**
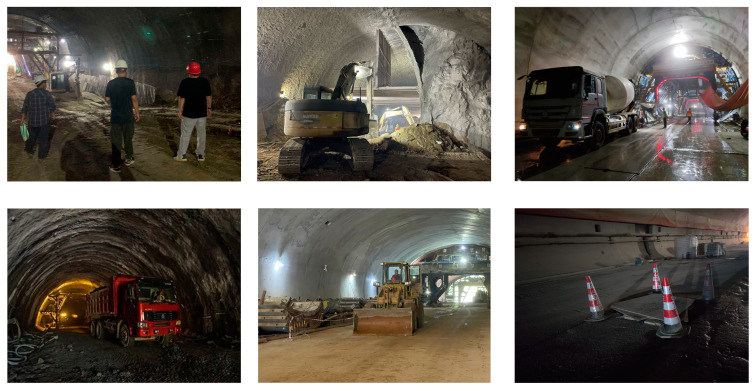
Representative images from different categories in dataset.

**Figure 17 sensors-24-00395-f017:**
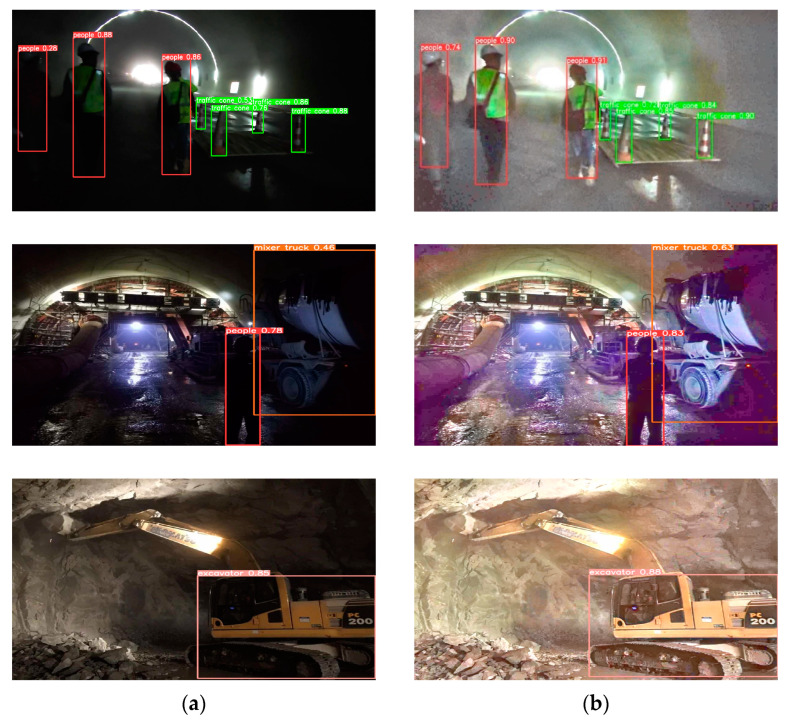
Low-light enhanced image. (**a**) Before low-light image enhancement; (**b**) after low-light image enhancement.

**Figure 18 sensors-24-00395-f018:**
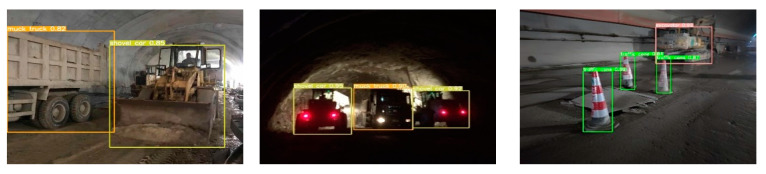
Improved model detection effect.

**Table 1 sensors-24-00395-t001:** Performance analysis of each model version of YOLOv5-6.0.

Model	mAP@0.5/%	Inference Speed/MS	Params/M
YOLOv5n	45.7	45	1.9
YOLOv5s	56.8	98	7.2
YOLOv5m	64.1	224	21.2
YOLOv5l	67.3	430	46.5
YOLOv5x	68.9	766	86.7

**Table 2 sensors-24-00395-t002:** Performance comparison of different backbone networks.

Network Structure	mAP@0.5/%	Inference Speed/MS	Params/M
Mobilenet v3	95.6	126.5	0.795
Shufflenet v2	89.9	49.5	0.219
Ghostnet	96.6	109.6	1.480

**Table 3 sensors-24-00395-t003:** Comparison of the improved YOLOv5 with different attention mechanisms.

Attention Mechanism	mAP@0.5/%	Inference Speed/MS	Params/M
SE	89.3	47.7	0.2
ECA	91	51.8	0.2
CBAM	90.3	48.7	0.2
CA	91.6	51.9	0.2
C3_CA	94.6	50.9	0.5

**Table 4 sensors-24-00395-t004:** Comparison of obstacle detection experimental results.

Model	mAP@0.5/%	Precision/%	Recall/%	Inference Speed /MS	Frame Rate /FPS	GFLOPs	Params/M	Model Size /MB
YOLOv7-tiny	97.6	95.3	94.6	149.2	6.7	13.1	6.02	12.3
YOLOv5n	96.7	95.2	94.4	98.7	10.3	4.2	1.7	3.9
Point 1	89.6	86.1	82.2	48.3	20.25	0.5	0.2	0.7
Point 2	94.6	91.3	88.8	44.6	21.8	0.8	0.5	1.4
Point 3	94.9	94.8	86.2	50.6	18.3	0.8	0.49	1.4
Point 4	95.2	94	88.8	61.6	15.9	0.8	0.5	1.4

## Data Availability

The data presented in this study are available on request from the corresponding author. The data are not publicly available as the dataset was independently constructed by our research team and has not been disclosed at this time.

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
