# Peer review of "Lightweight Tunnel Obstacle Detection Based on Improved YOLOv5"

_sensors, 2024, doi:10.3390/s24020395_

Round 1

Reviewer 1 Report

Comments and Suggestions for Authors

1, In the subsections of Section 4, it is only mentioned that the model size needs to be reduced for embedded development applications in industrial scenarios, so it is necessary to compare and replace modules. However, there may be a lack of specific descriptions regarding the actual problems in the field that led to the replacement of modules. The subsections after 4.2 lack these descriptions. It is recommended to provide more specific descriptions regarding the specific problems in the industrial scenario.

2, The abstract needs specific percentages for the accuracy decrease and speed increase, along with an analysis.

3, Figure 2 describing the architecture appears unclear.

4, The logic at the beginning of the "4. Model" section is problematic.

5, There are formatting issues with the units of performance indicators in all tables.

6, Performance comparison of different network models (Table 1) doesn't specify the dataset on which the performance metrics were obtained.

7, The comparison of attention mechanisms (Table 3) lacks information about the dataset and specific locations where attention mechanisms were added.

8, Each subsection in the entire "Model" section should include diagrams of the improved structural changes, such as modified backbone and neck structures.

9, Specific explanations are needed for module replacements, such as replacing Conv modules in the backbone, neck, and even within the C3 module.

10, All performance metrics mentioned in section 5.3 should be presented in the subsequent experimental results' performance comparison table.

11, The experimental results are not satisfactory; the performance of Point 2 is notably better than Point 4.

12, The experiments are too singular

Comments on the Quality of English Language

no comment

Reviewer 2 Report

Comments and Suggestions for Authors

In the Related Work section, the transition from the traditional approach to the current approach is suggested and the shortcomings of the traditional approach are pointed out. Further polishing is required.

Why is the YOLOv5 series used as the basic network instead of other networks?Why YOLOv5n? The introduction is not explained, and the logic of the introduction is not strong.

The Line 144 "The EnlightenGAN network for low-light enhancement" section suggests showing more image enhancement comparison images in different categories to illustrate the benefits of image enhancement for subsequent detection.

Lines 195-235. It is recommended that the original model be presented more concisely, with a focus on improving the model.

Is the data in Table 1 based on the dataset in this article?

Does Speed/MS in the table refer to FPS?

Line315. How does the C3_CA module combine C3 with CA, and where is the attention mechanism module such as SE added?

Line338. F_1([𝑧ℎ,𝑧𝑤]) does not correspond to the following formula

Line 415. It is recommended to place section 4.6 in a section below the original model for clearer logic

line 429 . Please provide the device used to collect the images, the resolution, the shooting height, etc., so that readers can have more in-depth inspiration for the research.

Figure 10 suggests adding the number of different categories, and the number of samples of different categories is not introduced in the text.

"5.1Dataset" suggests that more categories of image data be presented to improve understanding of the study.

The "5.3 Performance Metrics" recommendation is briefly introduced

How are the hyperparameters of Lines 457-462 set and why are they set up?

Line505 – 511 This section does not conduct a comparative test of the CBSM part, where Point1-4 is an experiment with the improvement of a single module, or an improvement on the previous one?

Line513 – 525 lacks analysis of test results. It is recommended to supplement the detection results under more categories, and compare them with the detection results of the original model and the detection effect with or without image enhancement.

Does the improved algorithm in this article actually work on embedded devices?

The "Conclusions" part is not logical enough, the description is too simplistic, and the results of the above improved methods are not discussed, which fails to reflect the advantages of the method in efficient detection in complex environments, and needs to be further modified.

Comments on the Quality of English Language

Minor editing of English language required

Reviewer 3 Report

Comments and Suggestions for Authors

This paper proposed a tunnel obstacle detection approach based on improved YOLOv5. This paper is relatively well written and clear. The flow and representation are good. Nevertheless, some improvements could be made before the publication. Some issues need to be addressed.

1.In the introduction, too little has been described about traditional obstacle detection methods, the advantages of the proposed approach cannot be accurately understood. Authors should provide further references.

2. The performance of different YOLO versions, Network Structures, and Attention Mechanisms are compared in the paper. Why are results with high detection accuracy sometimes chosen, and are results with high detection speed sometimes chosen? What is the impact of the chosen method on the final model? Which is more important in real-time performance and accuracy of algorithms in this paper?

3. In Dataset section, divided in an 8:1:1 ration, but Figure 10 illustrates the number of different categories varies greatly. In validation or test sets, whether existing enough “shovel car” and “mixer truck”. The data set in this paper is directly divided into training set, verification set and test set at a random ratio of 8:1:1, and the reasons should be highlighted.

4. How to deal with the problem of decreased accuracy in detecting objects after enhancing the original image with EnlightenGAN in Figure 11?

5. The description of the model and algorithm is relatively simple and lacks more detailed design details, such as the specific implementation of the attention mechanism and the algorithm flow of convolutional block modification.

6. The data set was only tested on a self-built tunnel obstacle data set, which was not verified in a wider and complex scenario, nor compared with other similar lightweight models, so the advantages and disadvantages of the model could not be evaluated more comprehensively.

7. In figure 11, the accuracy of each category detected by FIG. a and FIG. b after the original image was enhanced by EnlightenGAN did not improve significantly, and even the probability of some objects showed a downward trend.

Comments on the Quality of English Language

English can be improved.

Round 2

Reviewer 2 Report

Comments and Suggestions for Authors

No further issues.